# Screening of C. auris among Candida isolates from various tertiary care institutions in Lahore by VITEK 2 and real time PCR based molecular technique

**Zill-e- Huma**[1]*, **Sidrah Saleem**[1], **Muhammad Imran**[1], **Kokab Jabeen**[1], **Faiqa Arshad**[1], **Ali Amar**[2]

**1** Department of Microbiology, University of Health Sciences, Lahore, Pakistan, **2** Department of Human Genetics and Molecular Biology, University of Health Sciences, Lahore, Pakistan

* zhumairfan@gmail.com

## Abstract

*Candida auris* is a multidrug-resistant pathogen, that is a well-known cause of nosocomial infections. This pathogen is being identified using advanced diagnostic approaches and epidemiological typing procedures. In underdeveloped nations, several researchers developed and validated a low-cost approach for reliably identifying *Candida auris*. The goal of this study was to assess the burden of *Candida auris* in different teaching hospitals of Lahore and to limit its spread to minimize hospital-related illnesses. *Candida* isolates were obtained from various tertiary care institutions in Lahore in the form of culture on various culture plates. Sabouraud agar culture plates were used to culture the *Candida* spp. Fluconazole-resistant *Candida* species were chosen for further identification using VITEK 2 Compact ID and molecular identification using species-specific PCR assay. The current study obtained 636 *Candida* samples from several tertiary care institutions in Lahore. Fluconazole resistance was found in 248 (38.9%) of 636 *Candida* samples. No isolate was identified as *Candida auris* by VITEK 2 Compact ID and real-time PCR-based molecular identification. Thus with limited resources, these two methods may serve as useful screens for *Candida auris*. However, it should be screened all over the country to limit its spread to break the chain of nosocomial infections.

## Introduction

*Candida auris*, the first case of this emerging *Candida*, was reported in 2009 in Japan as an isolate from a clinal specimen of a patient's ear canal and was later isolated from various body sites of patients in numerous countries across the five continents [1]. *Candida auris* is rapidly expanding its clinical range around the world, from minor cases of superficial infections such as ear canal infections to extremely invasive cases such as bloodstream infections [2,3].

The problem of *Candida auris* epidemics in Europe appears to be growing, even though the epidemiologic details are not completely defined. Recently, the European Centre for Disease Control published an analysis of *Candida auris* testified cases and laboratory capacity to design

**Funding:** The authors received no specific funding for this work.

**Competing interests:** The authors have declared that no competing interests exist.

a surveillance mechanism and implement a control plan to prevent its spread [3,4]. Because of the underreporting of *Candida auris* and the imperfect accuracy of available conventional diagnostic tools, the actual prevalence of *Candida auris* remains unknown [5].

Despite intensive efforts to contain the spread of this rapidly spreading pathogen, several new cases have consistently emerged, with a proclivity to form an outbreak pattern [1]. *Candida auris* is the second most common *Candida* species responsible for invasive infection in India, necessitating more effective and functional infection control practices to prevent its spread [6]. The outbreak of infection by *Candida auris* in India was reported by Chowdhary detecting more than 10 cases between 2009 and 2012 [2]. More than 600 cases of *Candida auris* were reported in Europe between 2013 and four years ago [7]. The foremost *Candida auris* cases of invasive infection were reported in Spain, where four patients were diagnosed with *Candida auris* in deep-seated infections. Between April and June 2016, they were admitted to Valencia's Surgery unit's intensive care unit [8]. From April 2016 to January 2017, more than 100 patients were reported to be infected with *Candida auris*, with many developing candidemia and, most concerningly, some developing septic metastatic complications as a result of this *Candida*. The *Candida auris* outbreak is regarded as Europe's largest clonal outbreak, involving a new, previously unknown strain that was confirmed by genetic analysis [9].

The rising incidence of non-*albicans Candida* species colonization and infection in recent years is assumed to be largely caused by the increased use of prophylactic antifungal medications such as fluconazole [10]. Previously, *Candida albicans* was the primary cause of invasive candidiasis. Fluconazole is no longer the cornerstone of empirical antifungal treatment due to the shift toward non-*albicans Candida* species with varying susceptibility patterns, including multidrug-resistant species. *Candida auris*, given its proclivity to spread fast in critically ill individuals, has the potential to become a dominating opportunistic pathogen in these populations [1].

Molecular analysis has revealed a remarkable variance of *Candida auris* from the other *Candida* species, but it is phylogenetically correlated to the *C. haemulonii* species complex [11]. Previously, these organisms were rarely identified as the causative agents of invasive and deep-seated infection of bone and soft tissues. The infections are more severe in diabetics, immuno-compromised patients, and patients who have previously taken antifungal drugs [12,13]. The haploid genome of *Candida auris* is around 12.5 Mb, with a Guanine-Cytosine content (GC Content) of approximately 45 percent [14]. Genomic characterization proposes that there are about 6,500 and 8,500 protein-coding sequences, with a large number of genes that code for proteins encoding virulence factors [15].

In light of all the details regarding havoc and mayhem created by *Candida auris*, the present study was planned to rule out the unidentified or misidentified cases of *Candida auris* in Lahore, Pakistan to avoid its spread in the hospital setting. This study will also provide additional insight into the best method for screening *Candida auris* in limited resources by VITEK 2 ID and real-time PCR-based molecular identification.

## Materials and methods

It is a cross-sectional study, performed at the University of Health Sciences, Lahore. The study was performed after receiving approval from the Institutional Review Board (No: UHS/Immu/PhD/19-412), the Ethical Review Committee (No: UHS/REG-19/ERC/3525), and the Advanced Studies and Research Board (No: UHS/Education/126-19/4267) of the University of Health Sciences, Lahore, Pakistan, and also the institutional review committee (No: 2020-175-CHICH) of tertiary care hospital following the Declaration of Helsinki. For the present study, a total of 636 *Candida* isolates were collected during the study period from December

2019 to July 2021 from seven tertiary care hospitals (Mayo Hospital, Sir Ganga Ram Hospital, Services Hospital, Lahore General Hospital, Sheikh Zayed Hospital, and Children Hospital, Jinnah Hospital) of Lahore.

The sample size was calculated by using the formula given below using the Sample Size determination in health studies WHO version 2.0.21.

$$= \frac{Z^2 P(1-P)}{d^2}$$

Where $Z_{1-\alpha/2}$ is the critical value for a two-tailed test at 95% = 1.96, P = % of cases in previous study = 0.14%, and d = absolute precision = 0.05 calculates $n_o$ = 186.

By applying the finite population correction formula = $n = \frac{n_o}{1+\frac{(n_o-1)}{N}}$

N = population under study = 144

So the sample size is calculated as n = 81 $\approx$ 85 *Candida* species [16].

Cultures on various culture plates (blood agar culture plate, MacConkey agar, or Cysteine lactose electrolyte deficient (CLED) media agar plates) were used to transport the samples. The samples are given lab numbers and cultured on Sabouraud agar culture plates (Biolife, Italiana) for 24 hours at 35–37°C. The following day, the Sabouraud agar culture plates were examined for growth, and observations were made. If there was insufficient or no growth, the plates were incubated for another 24 hours at 37 C, and the results were observed the next day. Colonial morphology and microscopy were done to identify the *Candida* species.

## Antifungal susceptibility testing (AST)

The modified Kirby-Bauer disc diffusion method was used to test antifungal susceptibility. Mueller-Hinton Agar with 2% Glucose and 0.5 g/mL Methylene Blue Dye (GMB) Medium agar culture plate was uniformly seeded with the organism to be tested in this method [17,18]. All of the *Candida* species tested against commercially available antimicrobial disks (purchased and imported from Liofilchem, Italy) were used in this study. The Azole class of antifungal drugs is disregarded as an ideal treatment for *Candida auris* as the majority of strains are resistant to it. Fluconazole disc was used to screen the fluconazole-resistant *Candida* isolates. The results were interpreted using Clinical Laboratory Standard Institute (CLSI) M44 guidelines and the manufacturer's recommendations for the control (ATCC 10231) and test isolates of *Candida* species, which were labeled sensitive, intermediate, and resistant [19,20]. The samples were then identified using the VITEK 2 Compact system version 8.01. The VITEK 2 system (VITEK 2 Compact, Biomeriux, USA) is a fully automated system for identifying bacteria and yeasts. It has numerous advantages, including speed and ease of use with certainty.

## Identification of *Candida* species by VITEK 2 ID Compact system

In a sterile polystyrene tube, an inoculum of various *Candida* species was made from a 24-hour-old culture. The laboratory sample numbers were written on the polystyrene tubes. The inoculum was made by placing 3.0 ml of sterile normal saline in a polystyrene tube and adding 2–3 of the morphologically similar colonies. To achieve uniform turbidity of the inoculum, the tube was vortexed on the vortex mixer. The inoculum's McFarland turbidity was set to 2.0 McFarland. The cartridge was filled with a polystyrene tube. The VITEK 2 Identification card was removed from the sealed pack and transferred into the cartridge by carefully inserting its blue color-coded transfer tube into the tube while avoiding touching the tube's walls. It was decided that the cards should be set up within half an hour of the inoculum being diluted.

The VITEK 2 instrument system and computer were both turned on. The date of data entry, tests performed, laboratory sample number, presumptive identification, and accession

number were all entered into the system. All of the polystyrene tubes were installed in the cassettes alongside the VITEK cards. The cassette was placed in the VITEK 2's filler box. When the filler cycle was finished, the system beeped and the load door was unlocked. The cassettes were moved from the filler box to the load box. The VITEK cards were automatically filled and sealed under a vacuum. Following that, the instrument made another beep to remove the cassette and tubes. The instruments were loaded with the cards. The card processing was initiated. The results were read the following day. The control strain *Candida albicans* (ATCC 10231) was used for quality control.

### Identification of the *Candida auris* by qPCR

The DNA of the *Candida* samples was extracted by using the commercially available kit (Gene JET Genomic DNA purification kit, Thermoscientific, USA) according to manufacturer instructions. The primers for *Candida auris* were designed for the rapid identification of the *Candida auris* isolates, according to the primers and protocols as shown in Table 1, as described by [21]. Primers (forward and reverse) were synthesized from commercially available services (Synbio Technologies, USA) in a 25 nmol synthesis scale. These primers are known to target species-specific regions of *Candida auris*. The species specificity of the primers used in our study, CAURF and CAURR for *Candida auris* was indicated by BLAST searches as they exhibited comprehensive sequence homology with *Candida auris* strains only [21].

The PCR amplification was run on Rotor-Gene (Rotor-Gene Q, Germany). Master mix (Maxima SYBR Green/ROX q PCR, Maxima Hot Start Taq DNA Polymerase) was purchased from Thermo Scientific. The *Candida auris* RT-PCR amplification was performed under the following standard reaction conditions. The Maxima SYBR Green/ROX q PCR master mix: 15μl, Forward primer (10 μmol): 1 μl, Reverse primer (10μmol):1 μl, Nuclease Free Water: 6 μl and Template DNA (25ng/μl): 2 μl. This made a final volume of 25μl. The PCR was performed on Rotor-Gene (Rotor-Gene Q, Germany) under the thermal cycling profile as mentioned. PCR cycle was performed under conditions of one cycle at 95˚C for 5 min, followed by 30 cycles of 95˚C for 1 min, 52˚C for 30 s, and 72˚C for 1 min, followed by one cycle of final extension at 72˚C for 10 min as described by the study [21].

## Results

In this study, 636 *Candida* samples were collected from various tertiary care hospitals in Lahore The distribution of *Candida* isolates from different hospitals is described in Table 2.

The data regarding the age-wise distribution from different hospitals is shown in Table 3.

Fluconazole resistance was established in 248 isolates (38.9%) while the remaining 388 (61.0%) were sensitive to Fluconazole by disc diffusion method. The details of Fluconazole resistant isolates from various hospitals is shown in Table 4.

Table 5 shows the identification of various *Candida* species as determined by the VITEK 2 compact system. The remaining 13 samples run on VITEK 2 were unable to provide results due to the termination, so their results were not included here.

**Table 1. *Candida auris* design of forward and reverse primer.**

| Sr. No. | Primers | Sequence (5'- 3') | Length (bases) | Tm (˚C) | PCR product size (bp) |
|---|---|---|---|---|---|
| 1 | Forward primer | ATTTTGCATACACACTGATTTG | 22 | 51 | 276 |
| 2 | Reverse primer | CGTGCAAGCTGTAATTTTGTGA | 22 | 54 | |

**Table 2. Hospital-wise distribution of *Candida* isolates (n = 636).**

| Sr. No. | Hospital | Gender n (%) | | Total n (%) |
|---|---|---|---|---|
| | | Male | Female | |
| 1 | KEMC/ Mayo Hospital, Lahore | 97 (38.2) | 157 (61.8) | 254 (100) |
| 2 | The Children's Hospital and The Institute of Child Health, Lahore | 114 (56.7) | 87 (43.3) | 201(100) |
| 3 | FJMC/ Sir Ganga Ram Hospital, Lahore | 32 (31.4) | 70 (68.6) | 102 (100) |
| 4 | Sheikh Khalifa Bin Zayed Al Nahyan MDC/ SZH, Lahore | 16 (55.2) | 13 (44.8) | 29 (100) |
| 5 | SIMS/ Services Hospital, Lahore | 9 (42.9) | 12 (57.1) | 21 (100) |
| 6 | ADMC/ PGMI/ LGH, Lahore | 10 (47.6) | 11 (52.4) | 21 (100) |
| 7 | AIMC/ Jinnah Hospital, Lahore | 3 (37.5) | 5 (62.5) | 8 (100) |
| 8 | Total | 281 (44.2) | 355(55.8) | 636 (100) |

Fisher's Exact Test: 24.971 p-values: <0.001(highly significant).

## Molecular identification of *Candida auris*

The RT-PCR was performed on all of the selected isolates (n = 100) under the previously described conditions [21]. There was no amplification observed. Using the real-time PCR-based molecular method, all of the isolates were found to be negative for *Candida auris* with the specific set of primers used in this study.

**Table 3. Mean age comparison between hospitals using Post Hoc Test (LSD).**

| Hospitals | | Mean difference | p-value | Remarks |
|---|---|---|---|---|
| Children H. | Jinnah H. | -11.18 | 0.55 | Not significant |
| Children H. | LGH H. | -20.94 | 0.001 | Highly Significant |
| Children H. | Mayo H. | -15.36 | 0.001 | Highly Significant |
| Children H. | Services H. | -13.33 | 0.001 | Highly Significant |
| Children H. | SGR H. | -16.41 | 0.001 | Highly Significant |
| Children H. | SZ H. | -17.32 | 0.001 | Highly Significant |
| Jinnah H. | LGH H. | -9.76 | 0.145 | Not significant |
| Jinnah H. | Mayo H. | -4.18 | 0.470 | Not significant |
| Jinnah H. | Services H. | -2.14 | 0.749 | Not significant |
| Jinnah H. | SGR H. | -5.22 | 0.378 | Not significant |
| Jinnah H. | SZ H. | -6.14 | 0.341 | Not significant |
| LGH H. | Mayo H. | 5.58 | 0.128 | Not significant |
| LGH H. | Services H. | 7.61 | 0.126 | Not significant |
| LGH H. | SGR H. | 4.53 | 0.241 | Not significant |
| LGH H. | SZ H. | 3.62 | 0.433 | Not significant |
| Mayo H. | Services H. | 2.03 | 0.578 | Not significant |
| Mayo H. | SGR H. | -1.04 | 0.581 | Not significant |
| Mayo H. | SZ H. | -1.95 | 0.536 | Not significant |
| Services H. | SGR H. | 3.08 | 0.425 | Not significant |
| Services H. | SZ H. | -0.91 | 0.788 | Not significant |
| SGR H. | SZ H. | 0.91 | 0.788 | Not significant |

**Table 4. Fluconazole resistance in various hospital isolates (n = 636).**

| Sr No | Hospitals | Fluconazole Resistant | Fluconazole sensitive | Total |
|-------|-----------|-----------------------|-----------------------|-------|
| 1 | Children | 82 (12.8%) | 119(18.71%) | 201(31.60%) |
| 2 | Jinnah | 2(0.31%) | 6(0.94%) | 8(1.27%) |
| 3 | LGH | 6(0.94%) | 15(2.35%) | 21(3.31%) |
| 4 | Mayo | 110(17.29%) | 144(22.6%) | 254(39.93%) |
| 5 | Services | 10(1.57%) | 11(1.73%) | 21(3.31%) |
| 6 | SGRH | 33(5.19%) | 69(10.85%) | 102(16.03%) |
| 7 | SZH | 5(0.78%) | 24(3.77%) | 29(4.56%) |
|   |   | 248(38.99%) | 388(61.01%) | 636(100%) |

Likelihood Ratio-12.998 p-value-$<0.05$ (Significant).

## Discussion

*Candida auris* is regarded as a global threat for three reasons: it is a multidrug-resistant emerging fungus that is difficult to identify using standard microbiological procedures, resulting in inappropriate management, and it has a strong proclivity to cause outbreaks in hospital settings. *Candida auris* has been added to the CDC's list of nationally notifiable diseases, and confirmed clinical cases, as well as probable cases, should be reported to local health authorities and the CDC as soon as possible [22,23].

There was no confirmed and clear data regarding the presence of *Candida auris* cases in Lahore, Pakistan, at the start of the study. *Candida auris* was not detected in this study using the VITEK 2 compact identification system and qPCR. Both methods yielded no positive results for *Candida auris*. The author, Abd-Elmonsef, stated his findings about his research on the *Candida* species in a very recent study published in 2022. *Candida auris* was found in Tanta University Hospital in Egypt, according to the author. He used the VITEK 2 compact identification system to identify the isolated *Candida* species, and then he used the VITEK 2 compact YST08 to determine antifungal susceptibility testing of the isolated *Candida* species [24]. The researcher used PCR to perform molecular confirmation for the identification of *Candida auris*. All *Candida* species were genotyped and identified by the author. According to the author, no *Candida auris* was found using the VITEK 2 identification system or the polymerase chain reaction [24].

**Table 5. *Candida* species identified by VITEK 2 Compact.**

| Sr no | *Candida* Species identified by VITEK 2 | Gender | | Number isolated n (%) |
|-------|------------------------------------------|--------|---|------------------------|
|       |                                          | M | F |                        |
| 1 | *Candida glabrata* | 12 | 13 | 27(31.8) |
| 2 | *Candida albicans* | 10 | 12 | 22(25.9) |
| 3 | *Candida tropicalis* | 7 | 6 | 13(15.3) |
| 4 | *Candida famata* | 3 | 9 | 12(14.1) |
| 5 | *Candida krusei* | 5 | 1 | 6(7.1) |
| 6 | *Candida intermedia* | 1 | 1 | 2(2.4) |
| 7 | *Candida guilliermondii* | 1 | 1 | 2(2.4) |
| 8 | *Candida spherical* | 1 | 0 | 1(1.2) |
| 9 | *Candida parapsilosis* | 1 | 0 | 1(1.2) |
| 10 | *Candida catenulate* | 0 | 1 | 1(1.2) |
| 11 | Total | 41 | 46 | 87 |

This study's findings are consistent with our findings. VITEK 2 was used in the present study to identify *Candida* species through a variety of biochemical reactions included in VITEK 2 compact, and the results were confirmed by the polymerase chain reaction. It is recommended that no further testing be performed if the identification of *Candida auris* was performed by the VITEK 2 version 8.01, as the new updated software of VITEK 2 version 8.01 identification software has contained the taxon of *Candida auris* within it. In the present study, the VITEK 2 identification system has not detected a single strain of *Candida auris*. Furthermore, we used a polymerase chain reaction to double-check *Candida auris*. These primers have been shown to target specific sequences that are species-specific to the *Candida auris*. The RT-PCR technique also revealed no *Candida auris* isolates.

Ambaraghassi et al. conducted another study on *Candida auris*. According to the author, the phenotypic methods are frequently used in clinical microbiology laboratories to identify *Candida auris*. VITEK 2 YST ID was used by the author to identify *Candida auris* and related species. According to the author, the VITEK 2 automated identification system recently added *Candida auris* to its database in an updated version. The author also suggests that the isolation of *Candida duobushaemulonii* should trigger further testing to rule out *Candida auris* [25] We used the updated version of VITEK 2 to identify the *Candida* species in this study, and no isolate of *Candida auris* was found. Tan et al. demonstrated that the VITEK 2 system could correctly identify South Asian *Candida auris* clades, misidentify East Asian stains, and provide a low discrimination result for South American strains in a study [26]. This finding supports our study because it has been demonstrated that the VITEK 2 compact system can correctly identify the South Asian *Candida auris* clades and not a single isolate was identified as *Candida auris* by the VITEK 2 compact system.

Recently, Sana et al. reported on the success of managing a *Candida auris* outbreak in a tertiary care hospital in Rawalpindi, Pakistan. The index case was thought to be a 58-year-old woman who was admitted with stomach carcinoma and underwent multiple laparotomies. During her stay in the ICU, she developed sepsis and was treated with a variety of antifungal and antibacterial agents [27]. *Candida auris* was isolated from her blood cultures in July 2018. *Candida auris* was then isolated from eight more patients over four months. The *Candida* isolates were initially misidentified as *Candida famata* and *Candida hemulonii*. The cases were then identified as *Candida auris* using VITEK 2, rather than the molecular method [27]. The findings of Sana et al. are consistent with our findings because we used the most recent VITEK 2 identification software version 8.01 which is capable of correctly identifying *Candida* to the species level. Furthermore, we confirmed our findings by genetically identifying *Candida auris* by species-specific real-time PCR-based method, and not a single sample was identified as *Candida auris*.

*Candida auris* was found in the Agha Khan Hospital in Pakistan in another study by Sayeed et al. According to the study, in September 2014, Aga Khan University Hospital in Karachi, Pakistan, experienced an outbreak of a yeast infection primarily identified as *Saccharomyces cerevisiae*. A total of 92 yeast cases were discovered. Because this isolated yeast was found to have an unusual antifungal susceptibility pattern, it was decided to retest some of the yeast isolates at the Centers for Disease Control and Prevention (CDC), Atlanta, USA for final identification, and the unusual yeast was eventually identified as *Candida auris*. In D1-D2 sequencing, all 15 strains sent for whole genome sequencing show 100 percent concordance [28]. In contrast, in our study, we double-confirmed the *Candida auris*, one by VITEK 2 identification and the other by genetic identification by using a species-specific real-time PCR-based method, which was regarded as a low-cost alternative to the sequencing. It is also supported by a study in which a biochemical system and genetic identification are viewed as a low-cost and effective alternative to costly identification procedures such as Matrix-assisted

laser desorption ionization-time-of-flight mass spectrometry (MALDI-TOF MS) and sequencing [29].

## Conclusion

In the current study, no *Candida auris* isolate was identified using the VITEK 2 Compact identification system or real-time PCR-based molecular identification. Thus with limited resources, these two methods may serve as useful screening methods for *Candida auris* in the hospital setting.

## Limitations and future prospects

Furthermore, multiple studies should be planned across the country to detect this ruinous microorganism early and limit its spread. In the present study, all fluconazole-resistant samples should be run through VITEK 2 and qPCR should be performed. We are unable to do so due to financial constraints. If all samples were screened by MALDI-TOF-MS and methods were compared, much better identification and results could have been obtained.

## Supporting information

**S1 File.**
(XLSX)

## Author Contributions

**Conceptualization:** Zill-e- Huma, Sidrah Saleem.

**Data curation:** Zill-e- Huma, Ali Amar.

**Formal analysis:** Zill-e- Huma, Ali Amar.

**Investigation:** Zill-e- Huma.

**Methodology:** Zill-e- Huma.

**Project administration:** Muhammad Imran.

**Resources:** Zill-e- Huma.

**Software:** Kokab Jabeen.

**Supervision:** Sidrah Saleem, Muhammad Imran.

**Validation:** Faiqa Arshad.

**Writing – original draft:** Zill-e- Huma.

**Writing – review & editing:** Zill-e- Huma.

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
