## [Decision Letter · Decision Letter 0]

1 Aug 2023

PONE-D-23-13143Molecular Identification of Candida auris by VITEK 2 and ITS-1 and ITS-2 regions of rDNA to early detect and limit infection in hospital setting of Lahore, PakistanPLOS ONE

Dear Dr. Huma,

Thank you for submitting your manuscript to PLOS ONE. After careful consideration, we feel that it has merit but does not fully meet PLOS ONE’s publication criteria as it currently stands. Therefore, we invite you to submit a revised version of the manuscript that addresses the points raised during the review process.

We look forward to receiving your revised manuscript.

Kind regards,

Aijaz Ahmad, Ph.D.

Academic Editor

PLOS ONE

Journal Requirements:

a) The name of the colleague or the details of the professional service that edited your manuscript.

b) A copy of your manuscript showing your changes by either highlighting them or using track changes (uploaded as a *supporting information* file).

c) A clean copy of the edited manuscript (uploaded as the new *manuscript* file).

3. Thank you for stating the following financial disclosure: "no"

5. Please amend your authorship list in your manuscript file to include author "Ali Amar".

6. Please amend either the abstract on the online submission form (via Edit Submission) or the abstract in the manuscript so that they are identical.

7. We note you have included a table to which you do not refer in the text of your manuscript. Please ensure that you refer to Table 2 in your text; if accepted, production will need this reference to link the reader to the Table.

Reviewers' comments:

Reviewer's Responses to Questions

**Comments to the Author**

1. Is the manuscript technically sound, and do the data support the conclusions?

Reviewer #1: No

Reviewer #2: Partly

2. Has the statistical analysis been performed appropriately and rigorously? 

Reviewer #1: I Don't Know

Reviewer #2: No

3. Have the authors made all data underlying the findings in their manuscript fully available?

Reviewer #1: Yes

Reviewer #2: No

4. Is the manuscript presented in an intelligible fashion and written in standard English?

Reviewer #1: Yes

Reviewer #2: Yes

5. Review Comments to the Author

Reviewer #1: The draft "Molecular Identification of Candida auris by VITEK 2 and ITS-1 and ITS-2 regions of rDNA to early detect and limit infection in hospital setting of Lahore, Pakistan" can be important if author do major revision and make necessary additions to draft. My specific comments are

1) Draft need minor writing check

2) In support of table 2, author have not shown any data like drug sensitivity or tolerance by spotting isolates on drug plate

3) Dont know what author want to convey from RT-PCR data shown in figure, there should be a positive control in RT-PCR to show that PCR works

4) Experiments were done with poor logic and proper controls are missing

Reviewer #2: Reviewer Recommendation and Comments for Manuscript Number PONE-D-23-13143

Article Title:

Molecular Identification of Candida auris by VITEK 2 and ITS-1 and ITS-2 regions of rDNA to early detect and limit infection in hospital setting of Lahore, Pakistan

Comments & Recommendations:

The title of the article is not suitable and it should be revised because the author did not identify any single isolate as Candida auris by any technique. However, she tried to screen the C. auris in the samples. One of the proposed title in my opinion is, “Screening of C. auris among Candida isolates from various tertiary care institutions in Lahore by VITEK 2 and real time PCR based molecular technique”.

Methodology:

The author collected Candida isolates from different tertiary care institution of Lahore and screened them for the presence of C. auris by testing Fluconazole @ 25μg.

Comments & Recommendations:

It may be mentioned here that Minimum Inhibitory Concentration for C. auris is known to be 32μg, therefore if any one desired to screen on the basis of fluconazole resistance, then the used concentration should be appropriate. Moreover, the best antifungal activity against C. auris is known by the use of Echinocandins (Please refer Sanyaolau et al., 2022; https://doi.org/10.3947/ic.2022.0008). Therefore, the author is advised to screen Echinocandins rather than Fluconazole. More references may be found if properly searched.

Results:

The author mentioned that she screened 636 Candida isolates and for fluconazole resistance and found resistance in 248 (38.9%) isolates and identify 87 isolates as different Candida species on the basis of VITEK 2 ID Compact System and performed RT-PCR on selected isolates (n=100).

Comments & Recommendations:

The author did not mention the criteria for selecting 87 and 100 isolates rather than testing all 248 fluconaozole resistant Candida isolates. However, she is claiming that C. auris could not be identify from collected samples.

It is suggested that to author to:

• Screen isolates resistant isolates using echinocandins or any other or appropriate concentration of fluconazole antifungal.

• Test all resistant Candida isolates or all collected isolates using VITEK 2 ID Compact System and RT-PCR.

• The author is also advised to use any control sample (positive control) for comparing the amplification on RT-PCR.

Statistical Analysis: The author did not perform any statistical analysis.

Comments & Recommendations: The author is advised to:

• Mention the name of institutes form where the samples were collected with exact numbers and gender.

• The data may be classified into various institutes/hospitals and gender (male & female).

• The data may analyze statistically in terms of means, standard error and standard deviation.

Overall Comments & Recommendations:

The manuscript may be sent to author for major revision. The manuscript may be accepted after incorporating all the comments.

6. PLOS authors have the option to publish the peer review history of their article (what does this mean?). If published, this will include your full peer review and any attached files.

Reviewer #1: No

Reviewer #2: No

---

## [Author Response · Author response to Decision Letter 0]

3 Sep 2023

Dear Reviewers

We have accomplished all the required recommendations in research article.

In case of any query, we will be happy to response again. 

warms regards

Dr Zill-e-Huma

---

## [Decision Letter · Decision Letter 1]

12 Oct 2023

Screening of C. auris among Candida isolates from various tertiary care institutions in Lahore by VITEK 2 and real time PCR based molecular technique

PONE-D-23-13143R1

Dear Dr. Zill-e- Huma,

We’re pleased to inform you that your manuscript has been judged scientifically suitable for publication and will be formally accepted for publication once it meets all outstanding technical requirements.

Kind regards,

Aijaz Ahmad, Ph.D.

Academic Editor

PLOS ONE

Additional Editor Comments (optional):

Reviewers' comments:

Reviewer's Responses to Questions

**Comments to the Author**

1. If the authors have adequately addressed your comments raised in a previous round of review and you feel that this manuscript is now acceptable for publication, you may indicate that here to bypass the “Comments to the Author” section, enter your conflict of interest statement in the “Confidential to Editor” section, and submit your "Accept" recommendation.

Reviewer #1: All comments have been addressed

2. Is the manuscript technically sound, and do the data support the conclusions?

Reviewer #1: Yes

3. Has the statistical analysis been performed appropriately and rigorously? 

Reviewer #1: Yes

4. Have the authors made all data underlying the findings in their manuscript fully available?

Reviewer #1: Yes

5. Is the manuscript presented in an intelligible fashion and written in standard English?

Reviewer #1: Yes

6. Review Comments to the Author

Reviewer #1: Revised manuscript looks OK for acceptance. Authors were able to addressed to the comments appropriately.

7. PLOS authors have the option to publish the peer review history of their article (what does this mean?). If published, this will include your full peer review and any attached files.

Reviewer #1: **Yes: **RAVINDER KUMAR

Reviewer #2: No

---

## [Editor Report · Acceptance letter]

16 Oct 2023

PONE-D-23-13143R1 

*Screening of C. auris among Candida isolates from various tertiary care institutions in Lahore by* VITEK *2 and real time PCR based molecular technique*

Dear Dr. Huma:

I'm pleased to inform you that your manuscript has been deemed suitable for publication in PLOS ONE. Congratulations! Your manuscript is now with our production department. 

Kind regards, 

on behalf of

Dr. Aijaz Ahmad 

Academic Editor

PLOS ONE